# Vegetarian Diet and Dietary Intake, Health, and Nutritional Status in Infants, Children, and Adolescents: A Systematic Review

**DOI:** 10.3390/nu17132183

**Published:** 2025-06-30

**Authors:** Daniela Reis, Melanie Schwermer, Lara Nowak, Nibras Naami, Tycho Jan Zuzak, Alfred Längler

**Affiliations:** 1Institute for integrative Medicine, Professorship for Integrative Pediatrics, Witten/Herdecke University, 58448 Witten, Germany; m.schwermer@gemeinschaftskrankenhaus.de (M.S.); a.laengler@gemeinschaftskrankenhaus.de (A.L.); 2Department of Pediatrics, Gemeinschaftskrankenhaus Herdecke gGmbH, Gerhard-Kienle-Weg 4, 58313 Herdecke, Germany; lara.nowak@gemeinschaftskrankenhaus.de (L.N.); n.naami@gemeinschaftskrankenhaus.de (N.N.); t.zuzak@gemeinschaftskrankenhaus.de (T.J.Z.)

**Keywords:** vegetarian diet, children, adolescents, infants, health, dietary intake, nutritional status

## Abstract

**Background/Objectives**: More children and adolescents are adopting vegetarian diets. A balanced diet is particularly important for growth and bone development in children. It is important to clarify whether a vegetarian diet affects these processes. A systematic literature review was conducted to identify studies investigating differences in anthropometric data, nutrient intake, and biomarkers between vegetarian and omnivorous children. **Methods**: PUBMED, MEDLINE, and Web of Science Core Collections were searched between the end of 2014 and 2023. We included peer-reviewed randomized controlled trials, intervention, or observational studies that were published in English or German and investigated the differences between healthy children and adolescents from high-income countries who consumed either a vegetarian or an omnivorous diet. The review was conducted in accordance with the PRISMA guidelines. If at least five values with the same unit were available from different studies, a cumulative analysis of selected parameters was conducted. Due to the participants’ varying ages across the studies, limited cumulative analyses were conducted additionally by age category. **Results**: A total of 1681 studies were screened, of which 20 met the inclusion criteria. Significant differences were found in fiber and energy intake from carbohydrates and proteins. The results were strengthened by the restricted cumulative analysis of the 2–10 age category, which also revealed significant differences when comparing VG and OM. **Conclusions**: A higher intake of fiber, more energy from carbohydrates, and sufficient energy from proteins and less from fat can be regarded as the benefits of a vegetarian diet, according to this review. There are also the first indications of enhanced vitamin C and E, iron, folate, and magnesium intake. These characteristics can be regarded as potential benefits of a vegetarian diet. A lower vitamin B12 and vitamin D intake has been identified as a potential risk factor. Further longitudinal, prospective, observational studies are needed. Prospero registration date and number: 6 March 2023, CRD42023402301.

## 1. Introduction

The number of individuals worldwide who have adopted a vegetarian diet (VG), which excludes meat, fish, and occasionally, also eggs and milk, is increasing [1]. It is challenging to find reliable data on the prevalence. In selected European countries, the prevalence of VGs among adults ranges from 2 to 9% [2,3,4,5,6]. Data from the German Health Interview and Examination Survey for Adults (DEGS1) and the German Health Interview and Examination Survey for Children and Adolescents (KiGGS) show that 4.3% of adult Germans (18–79 years) and 3.4% of children and adolescents (1.5% aged 6–11 years; 5.1% aged 12–17 years) follow a VG diet [6,7]. The association between VG diets and healthier body weights, lower cancer incidence, and reduced cardiovascular disease risk among adults is well-documented [8,9]. A possible risk of a VG diet is the development of nutritional deficiencies, such as vitamin B12 and iron deficiencies, due to a lack of natural sources [10,11,12]. Adequate supplementation is necessary for the benefits to outweigh the risks [13]. Especially, growing children and adolescents have increased energy and nutrient requirements [1,14]. Therefore, they are at a higher risk of inadequate nutrient intake, which is crucial for growth [14]. Some professional societies have published recommendation and statement papers regarding the use of a VG diet [15,16]. A recurring theme in these documents is the potential for nutrient deficiencies, most notably vitamin B12, in infants. They recommend the monitoring of physical development and dietary intake by a pediatrician [17,18]. This prompts the question of whether a diet that reduces or eliminates animal products is appropriate for children and adolescents.

The objective of this systematic review was to reflect on the current state of research on the positive and negative effects of a VG diet compared to an omnivorous (OM) diet in infants, children, and adolescents. Studies from 2014 to 2023 that analyzed the food consumption, energy and nutrient intake, and vitamin and mineral intake of VG and OM children from high-income countries, as well as their health status (e.g., anthropometry, vitamin and mineral status, and biomarkers), were screened.

## 2. Materials and Methods

### 2.1. Subsection Reporting and Registration

This systematic review was prepared in accordance with the Preferred Reporting Elements for Systematic Reviews and Meta-Analyses (PRISMA) [19]. The protocol was pre-registered in the International Prospective Register of Systematic Reviews (PROSPERO) [CRD42023402301].

### 2.2. Information Sources and Search Strategy

The PubMED literature database was searched using the Advanced Search Builder (https://pubmed.ncbi.nlm.nih.gov/advanced/) and MEDLINE was searched using LIVIVO (www.livivo.de). All databases and search engines were last accessed on 1 January 2024. The terms in PUBMED were searched in the entire text (all fields). The search terms used were (vegetarian OR vegetarian* OR vegan OR vegan*) AND (infant OR infant* OR infancy OR child OR child* OR toddler OR toddler* OR teen OR teens OR youth OR youth* OR young people OR adolescent OR adolescent* OR adolescence OR Kid OR kids OR Juvenil OR Juvenil* OR minor OR minor* Or Youngster OR Youngster* OR breastfeeding OR breastfed). All publications from the last ten years (2013–2023) were included. As the last systematic review on this topic included studies up to 6 November 2014, these studies were excluded from this review [1]. In Medline via LIVIVO, the search terms vegetarian AND child were searched in the title and studies from 7 November 2014 to 2023 were included. In Web of Science Core Collections, the keywords vegetarian AND child were searched in the abstract with the above time exclusion. In the next step, the literature lists of the included studies were also searched for additional studies in the specified time interval that we had not previously included.

### 2.3. Eligibility Criteria and Study Selection

We included randomized controlled trials (RCTs) and intervention or observational studies (cross-sectional or prospective, peer-reviewed) investigating dietary intake, biomarkers, or other health-relevant parameters in VG and OM infants, children, and adolescents (0–18 years) from high-income countries. We also included studies in which the VG group comprises children following a VG or VE diet. Systematic reviews, reviews, position papers, and meta-analyses were excluded. The following criteria also caused an exclusion: unclear definition of VG and VE diets, studies about macrobiotic diet, and studies about exclusively breastfed infants of VG mothers.

### 2.4. Selection Process and Data Collection

AL and DR reviewed the titles and abstracts of the articles filtered from the databases. The reviewers worked independently and were blinded to each other’s selection process. If the inclusion and exclusion criteria were unclear from the abstract, they searched the full text. In case of ambiguity regarding inclusion or exclusion, consensus was reached. DR extracted the following data from the selected articles: title; author; year of publication and data collection period; objectives of the study; recruitment of study participants; country where the study was conducted; study design; dietary assessment method; participant characteristics (number, sex, age, and type of diet); assessed parameters (energy and nutrient intakes, food group intakes, nutritional status, and biomarkers); main results; and conclusions.

### 2.5. Study Risk of Bias Assessment

The quality of the studies was assessed on the basis of the validity of the results, the size of the effect, the precision of the estimate, the transferability to other situations, and the applicability of the results [20]. In addition, the Newcastle–Ottawa Scale (NOS) was used to assess the risk of bias for each included study [21] (Table 1; for a comprehensive overview of the quality assessment, please refer to Appendix A).

### 2.6. Statistics

A cumulative analysis was performed for parameters with a minimum of five mean values with the same unit that were available for children in the VG and OM groups. The mean values from the studies were depicted in scatter plots. The cumulative mean value of all studies was indicated by a line with error bars denoting the standard deviation. An unpaired *t*-test was used to determine statistical significance. The generation of graphs and statistical analysis was facilitated by using GraphPad5 (GraphPad Software, Inc., Boston, MA, USA). Due to the heterogeneous ages of the study participants, we conducted a limited cumulative analysis of stratification by age category. We focused on the 2–10 age group in the cumulative analysis, excluding studies that examined participants of different ages. The majority of the excluded studies focused on adolescents.

## 3. Results

### 3.1. Study Selection

In total 96 reports were assessed for eligibility. Thirty-three of these records did not clearly distinguish between VG and OM diets. Thirteen reports did not address children or adolescents, and four reports were not from the relevant time period or not written in English or German. Seventeen reports were not from high-income countries. Two reports were duplicates. Seven previously included reports were subsequently excluded. In six reports, dietary type was only a covariate [42,43,44,45,46,47]. In one report, participants had disease-related risk factors [48]. Twenty articles with fifteen samples fulfilled the search criteria. The result of the data identification, screening, and selection process is shown in Figure 1.

### 3.2. Study Characteristics

Of the twenty articles (Table 1), sixteen articles were from Europe, of which nine articles were from Poland [24,25,26,27,28,29,30,34,36] and five articles from Germany [22,23,39,40,41], with the remaining articles from Finland and the Czech Republic [33,38]. Outside of Europe, one article was from the United States [37], one from Australia [32], one from New Zealand [35], and one from Canada [31]. With the exception of one study with a longitudinal cohort design [31], all other studies were cross-sectional. The two German articles by Alexy et al. are subanalyses of the same cohort. This cohort includes a VG, a vegan (VE), and an OM group [22,23]. In their three publications, Weder et al. also used the same cohort, which included a VG, a VE, and an OM group [39,40,41].

The total cohort size (VG and OM) ranged from 40 [37] to 8907 [31]. Two articles included Seventh-Day Adventists in whole or in part [32,37]. A VE group was evaluated separately in eight articles [22,23,30,33,38,39,40,41]. The age range of the articles included infants and children (6 month–3 years) [31,39,40,41], toddlers and children (over 1–10 years) [24,25,26,27,28,29,30,33,36], adolescents [32,35], or infants, children, and adolescents [22,23,34,37,38]. The control groups were either OM children and adolescents [22,23,24,25,26,27,28,29,30,33,34,36,38,39,40,41] or described as “non-VG” [31,32,35,37].

Anthropometric data were reported in 19 articles (17 cohort samples) [22,23,26,27,28,29,30,31,32,33,34,35,36,37,38,39,41,49,50]. Five articles (four cohort samples) from Germany, the USA, and New Zealand reported data on food consumption [23,35,37,39,40], while the two articles with the same cohort reported only differences in estimated breast milk intake [39,40]. An assessment of biomarkers was the subject of 13 studies: carbohydrate–fat metabolism [26,27,28], iron and hemoglobin status [22,24,30,31,33], blood lipids [22,30,31,32,33] markers of bone metabolism [22,25,27,29,30,31,33], and various vitamins, especially vitamin B12 status [22,30,33,38], were examined. A single article discusses differences in the biomarkers of oxidative/antioxidative stress [36]. Some form of energy or nutrient intake [22,23,24,26,27,28,29,30,33,35,36,37,39,40], vitamins [22,24,25,27,28,29,30,33,36,37,40], and/or minerals [22,24,25,27,28,29,30,33,37,40] was analyzed in 15 articles (13 cohort samples) (Table 1).

All of the 20 articles used dietary assessments, either dietary recalls weighted [22,23,38,39,40,41], estimated [24,25,26,27,28,29,30,33,36], repeated 24-h dietary recalls [35], food frequency questionnaires [50,51,52], or other standardized questionnaires, such as the Nutrition and Health Questionnaire and NutriSTEP [31].

### 3.3. Summary of Study Results

#### 3.3.1. Anthropometry

BMI values (reported with varying units) were calculated for 16 study cohorts, three of which showed significantly lower BMI values (Z-score in two studies) in the VG group than in the OM group [34,35,37]. Three of the sixteen study cohorts also reported that VG children were significantly shorter than their peers [30,31,34], including weak evidence in Elliott et al. [31], who also reported a higher chance of underweight for VG. The average BMI (kg/m^2^), height (cm), and weight (kg) of the VG and OM groups did not differ significantly across all studies (Figure 2A–C). These results are confirmed by the limited cumulative analysis (Figure 2D–F). The values used for the cumulative analysis of the respective parameters, as well as the detailed cumulative analysis itself, can be found in Appendix A and for the limited cumulative analysis (Appendix A). Ambroszkiewicz et al. found a significantly lower fat mass in VG children than in OM children in three of their study cohorts [26,27,28]. Nieczuja-Dwojacka et al. [34] also determined significant slimmer skin folds (under the shoulder, on the abdomen, and on the arm) in VG children (Table 1).

#### 3.3.2. Biomarkers

##### Nutritional Status (Vitamins, Minerals Including Iron)

The investigation of iron status has been the subject of several studies [22,24,30,31,33]. Markers of iron metabolism, such as hemoglobin, ferritin, hepcidin, and soluble transferrin receptor (sTfR), were analyzed. In three of these studies, the VG group had significantly lower ferritin levels (an indicator of the body’s iron stores) than the OM group [22,24,30]. Ambroszkiewcz et al. examined hepcidin and sTfR as novel markers of iron metabolism. They found reduced sTfR levels and increased hepcidin levels in VG children compared with OM children, which may indicate reduced iron reserves in VGs [24] (Table 1). Appendix A provides an overview of the potential differences in selected biomarkers, including ferritin, between children with VG and an OM diet.

Three studies reported significant differences in vitamin B12 status between VG and OM [22,30,38] groups (Table 1 and Appendix A for differences VG and OM). Alexy et al. reported that VG children had lower levels of holotranscobalamin, the metabolically active form of vitamin B12, and higher levels of methylmalonic acid, a marker of vitamin B12 deficiency, than OM children [22]. In contrast to these findings, another study showed that VG children had higher serum vitamin B12 (cyanocobalamin) levels than OM children and found a significant difference between VG, VE, and OM groups [38] (Table 1). Most VG children were vitamin B12-supplemented. Non-supplemented VGs had significantly lower B12 concentrations than OMs in one study [30].

##### Inflammation Markers

*C*-reactive protein levels (CRP), which are a marker of inflammation in the body, were significantly reduced in one of the four VG study groups [24]. Data from the studies by Ambroszkiewicz et al. suggest that the VG diet positively affects the adipokine and inflammatory profile due to a higher ratio of anti-inflammatory to pro-inflammatory adipokines and due to reduced serum leptin levels [26,27,28] (Table 1 and Appendix A).

##### Blood Lipids

Five studies analyzed blood lipids associated with cardiovascular risk factors, such as high-density lipoprotein (HDL), low-density lipoprotein (LDL), total cholesterol, and triglycerides [22,30,31,32,33]. A reduction in LDL and total cholesterol levels in VG children was demonstrated in the study by Grant et al., and in another study by Desmond et al., a reduction in HDL cholesterol and an increase in triglyceride levels in VG children were observed (Table 1). When cow’s milk was consumed, VG children had significantly higher levels of lipoprotein cholesterol, low-density lipoprotein cholesterol, and total cholesterol than VG children who did not [31]. See also Appendix A for an overview of potential differences between VG and OM groups.

##### Bone Metabolism

Significant differences in the markers of bone metabolism were found in two studies by Ambroszkiewicz et al. [25,29]. Markers such as osteoprotegerin, nuclear factor-B ligand, sclerostin, and Dickkopf-related protein 1 were similar between VG and OM children. Two markers, the bone alkaline phosphatase and the *C*-terminal telopeptide of type I collagen, were found to be significantly higher in VG children than in OM children. These results suggest an increased bone turnover rate in VG children (Table 1).

#### 3.3.3. Energy and Nutrient Intake

The total energy intake of VG and OM children was reported for 12 study cohorts [22,24,26,27,28,29,30,33,35,36,37,39]. The study results and the cumulative analysis (see Appendix A for values of all respective parameters and the analysis itself) demonstrated that there was no significant difference between dietary groups (Table 1, Appendix A; Figure 3A). A significantly lower protein intake of VGs compared to OMs was observed in five of nine studies [26,27,29,35,40]. In two studies, only a significant difference between the VE, VG, and OM groups was described [22,30]. The mean protein intake was found to be non-significantly different between the VG (60.83 g/d ± 11.45) and the OM groups (48.75 g/d ± 15.18) (Figure 4C, Appendix A). In contrast to daily protein intake, nine of ten studies [24,25,26,27,28,29,35,36,40] and the cumulative analysis showed that the percentage of energy derived from protein was significantly lower in VG children (12.16% ± 1.16 vs. 14.53% ± 1.19; *p* = 0.0003) (Table 1, Appendix A; Figure 3C).

In one of four study cohorts [37], total fat intake was reported to be significantly lower in VG children and a reduced energy intake derived from fat was shown in another study [27]. A cumulative analysis of fat intake (61.31 g/d ± 11.25 vs. 68.22 g/d ± 9.71) and of energy derived from fat (32.14% ± 2.57 vs. 33.61 ± 1.86) also revealed no significant differences between groups (Figure 3D and Figure 4D; Appendix A). Two studies reported the cholesterol intake [33,40]. In both studies, VG had lower cholesterol intake than OM. In one study, this difference was significant [33]. The consumption of carbohydrate only differed significantly between VG and OM children in one of the five study cohorts [37]. Seven of the ten studies found that VG children had a significantly higher energy intake from carbohydrates (Table 1, Appendix A). In the study by Alexy et al., 2021, the significant discrepancy refers to all analyzed dietary groups, VE, VG, and OM [22]. The cumulative analysis did also not reveal any significant differences in daily carbohydrate intake (236.1 g/d ± 23,89 vs. 222.5 g/d ±15.76), but the analysis elicited a significant higher percentage amount of energy from carbohydrates in the VG group (54.57% ± 4.107 vs. 51.11% ± 2.215; *p* = 0.0307) (Figure 3B and Figure 4B; Appendix A). In contrast to protein, carbohydrates, and fat intake, fiber intake showed an obvious difference. Ten studies [22,26,27,29,30,33,35,36,37,39] and the cumulative analysis demonstrated that VG children (22.20 g/d ± 4.48) consumed significantly (*p* = 0.01) more fiber than OM children (16.67 g/d ± 3.48) (Table 1, Appendix A; Figure 4A). The results for energy from protein and carbohydrates were confirmed in the limited cumulative analysis, and there was also a significant difference in energy from fat and total energy intake (Figure 3E–H; Appendix A). The significant difference in fiber intake was confirmed in the limited analysis. There was also a significant difference in fat and protein intake in the limited cumulative analysis, but only five or three studies were included (Appendix A).

#### 3.3.4. Intake of Micronutrients

Nine studies [22,25,27,28,29,30,33,37,40] reported calcium intake (Table 1). A significant decrease in calcium intake was observed in the VG group compared to the OM group in one study [29]. VG diet was found to have significantly higher calcium intake compared to OM in another study [37]. Additionally, Desmond et al. reported a significant difference between the VE, VG, and OM groups, whereby calcium intake was increased in VGs in comparison to OMs [30]. Eight studies reported magnesium intake [22,27,28,29,30,33,37,40]. Studies by Segovia-Siapco et al., Hovinen et al., and Weder et al. reported a significant increase in magnesium intake for VGs compared to OMs [33,37,40]. In another study [30], a significant difference was observed between the VE, VG, and OM groups, with higher magnesium intake reported for VEs and VGs. A significantly higher iron intake in VGs was observed in four of six study cohorts [22,24,30,33,37,40]. The mean daily intake of calcium, magnesium, and iron in milligrams did not differ significantly between VG and OM children across all studies reporting data on these nutrients (Figure 5A–C; Appendix A for values of all respective parameters and the analysis itself). Furthermore, for the minerals manganese and potassium, a higher intake in VGs was shown in the study of Ambroszkiewicz et al. [28] and Segovia et al. [37], respectively. A reduced zinc intake in VGs compared to OMs was found in one study, while Hovinen et al. reported a higher zinc intake in VGs compared to OMs [33,37].

These results are confirmed by the limited cumulative analysis (age group 2–10) (Figure 5D–F, Appendix A).

#### 3.3.5. Intake of Vitamins

Vitamin C and vitamin E intake tend to be higher in VG children than in children with an OM diet. A total of eight studies reported on the intake of vitamin C [22,24,28,30,33,36,37,40]. Three of these studies reported significantly higher vitamin C intake in VGs compared to OMs [24,28,37]. One study demonstrated a significantly higher vitamin E intake in VGs than in OMs [37].

Conversely, vitamin B12 and vitamin D intake tend to be lower in the VG group than in the OM group. Significantly lower vitamin B12 intake was reported for VGs in four [22,25,28,36] out of ten study cohorts [22,25,27,28,29,30,33,36,37,40]. Significantly lower vitamin D intake for VGs was observed in one study [29], and a further study reported about a significant difference in vitamin B12 and D intake between OM, VE, and VG groups without supplementation, with lower values in VG and VE [30]. Cumulative analysis further revealed no significant difference in the intake of vitamins B12, C, D, and E between VG and OM children (Figure 5A–D and Appendix A for values of all respective parameters and the analysis itself). These results are only confirmed for vitamins C, D, and E by the limited cumulative analysis (age group 2–10) (Figure 6E–H and Appendix A).

#### 3.3.6. Food Groups

Alexy et al. [23], Segovia et al. [37], and Peddie et al. [35] reported on food group consumption in VG and OM children using different measurement methods (e.g., grams/MJ; the average daily nutrient intake; and percentage of OM and VG consumers). The study cohort of Segovia-Siapco et al. observed a significantly higher consumption of vegetables and fruits by VGs [37]. Cereals, cereal products, and carbohydrate-rich foods were consumed more frequently by VGs than by OMs [23,37]. Milk or dairy products were consumed significantly less by VGs than by OMs in two studies [23,37]. Sugar-sweetened beverages were consumed significantly more often by OMs than by VGs [37]. A significantly higher percentage of VGs consumed legumes, nuts, and meat alternatives than OMs did [35,37] (Table 1).

## 4. Discussion

### 4.1. Anthropometry

A balanced and adequate diet is essential for growth and bone development. The data on BMI, height, and weight found here show no significant differences between children who eat a VG diet and those who do not. The results were strengthened by the restricted cumulative analysis of the 2–10 age category. However, significant differences in BMI were mostly found in cohorts that included children over the age of 10 [35,37]. One of the cohorts in these studies that reported significant differences in BMI only included girls [35]. Of the three studies that reported significant size differences, one found VGs to be underweight [31]. Of the three studies that reported significant BMI differences, the VG children were slimmer [34,35,37]. However, the cumulative analyses in this review showed that the BMIs of VG and OM were not significantly different. Elevated BMI and excess body fat are both associated with an increased risk of metabolic disease in children and adolescents [49]. Therefore, a more favorable BMI and lower fat mass are preferable. Regarding height, the significantly smaller height of VGs compared to OMs in one study was within height norms [34]. Cumulative analysis showed that differences in height among the other studies were not significant. It remains unclear whether any height differences persist into adulthood. Reduced energy intake, or a combination of energy and nutrient deficiencies, has been discussed as a cause of height differences between children and adolescents on VG or VE diets and children and adolescents on OM diets [14,50]. The consumption of cow’s milk [51] and the consumption of animal protein compared to vegetable protein have also been cited as causes of height differences [53]. However, animal protein is often ingested through milk. It has been discussed that milk may influence growth not only through its nutrient content but also through its IGF1 content. The influence of IGF1 from animal and plant proteins on growth has not been conclusively clarified [54]. The final question is whether a maximum height is actually healthy or what normal healthy growth actually means [52].

### 4.2. Nutrient Intake and Biomarker

#### 4.2.1. Health Risk Markers (Inflammation Markers, Bone Markers)

In adults, a VG or VE diet leads to a better ratio of anti-inflammatory to pro-inflammatory markers and a lower risk of atherosclerotic burden and cardiovascular disease [55,56,57]. Atherosclerotic vascular changes begin in childhood [58]. Bone health in terms of bone density is considered critical, especially in VEs [59]. One reason could be the lower BMI, especially in VEs, which has a negative effect on bone density [60]. Elevated Alkaline Phosphatase (ALP) levels have been reported as a possible cause of increased bone turnover in adults with VG and VE [61]. One study included in our review also confirms that VGs have an increased BALP level compared to OMs [25].

#### 4.2.2. Energy and Nutrient Intake

The review shows that there is no significant difference in daily calorie intake between OM and VG children. This result was not supported by the restricted cumulative analysis of the 2–10 age category. It is possible that underlying factors are associated with age-specific variations in total energy intake. There appears to be little difference in their nutrient intake. Due to the small number of studies, the significantly different protein and fat intake identified in the limited analysis should be interpreted with caution. One exception is fiber. Here, it can be seen that VG children consume larger quantities. Additionally, it appears that VG children appear to absorb more energy from carbohydrates than from proteins. Due to their lower protein and fat intake, the carbohydrate intake of VG and especially VE was higher than that of OM. The quality of the carbohydrates was also better due to the lower intake of added/free sugars in the plant-based diets and the higher fiber intake. In adults, an increased intake of carbohydrate and fiber as part of a plant-based, low-fat diet has been shown to improve body weight, body composition, and insulin resistance [62]. Similar effects in children are possible but have not been studied.

#### 4.2.3. Protein Intake and Amino Acid Status

The daily amount of protein intake and the energy derived from protein tended to be higher in OM children. The reference values indicated that the protein intake in VG was sufficient, whereas the OM protein intake clearly exceeded the reference values. However, the quality of the protein (PDCAAS) in VG is controversial due to the pattern of amino acids. The PDCAAS of most plant proteins—such as cereals, legumes, and nuts—is lower than that of animal proteins, except for soy protein. However, Mariotta et al. point out that plant proteins are consumed in combination, which improves the protein quality of plant-based diets [63]. Legumes, nuts, and seeds provide adequate protein and amino acid intake. However, adequate energy intake is a prerequisite [63], which was observed in the studies included in this review. Unfortunately, studies evaluating amino acid intake in VG and VN diets in children are still lacking. In contrast, intakes of other nutrients, especially carbohydrates and their quality, magnesium, vitamin C, vitamin E, and folate were higher in the VG and VE diets.

#### 4.2.4. Fat, Cholesterol Intake, and Status

Some included studies showed a higher fat intake of OM compared to VGs or VEs who had the lowest fat intake. In the limited cumulative analysis, the energy from fat differed significantly. However, the fat quality, i.e., the proportions of saturated and unsaturated fat, is better in VGs and VEs than in OMs, as meat and milk (products) in a mixed diet contribute to the intake of saturated fatty acids [8]. However, fish, as the most important source of preformed long-chain *n*-3 fatty acids, is lacking in a VE diet. It may also be lacking in a VG diet. In a VG or VE diet, hardly any EPA or DHA is consumed, and the conversion of the n3 fatty acid alpha-linolenic acid is limited [16,64]. Single-cell oil-enriched foods or supplements are also good sources of DHA and EPA in a VG and VE diet [65]. However, it is unclear how often these are used. Studies using the biomarkers of fatty acid status in children on different diets would be important. Two of the included studies showed a higher cholesterol intake in OM diets compared to VG diets. These results are also confirmed in other studies [66]. In only one study in our review was the HDL level significantly lower in VG than in OM [30]. The results for LDL were inconsistent when subgroup analyses were not performed. The results may also depend on whether the VG group consumed milk or not.

#### 4.2.5. Vitamin and Mineral Intake and Status

This review confirms lower intakes of vitamins B12 and D in VGs. Because animal-based foods are the only reliable food source of vitamin B12, this vitamin is of particular interest in the evaluation of plant-based diets in children. As VGs have a lower consumption of animal foods (no meat, fish, less dairy products), vitamin B12 intake is lower in this group, as shown in this review. When assessing B12 intake in studies, it is important to take into account possible supplementation, which is particularly recommended for VEs [67]. Koller et al. found no significant differences in vitamin B12 intake [68]. However, the frequency of supplementation varies between countries, partly due to different healthcare systems and cultural differences. Since vitamin D improves calcium absorption and has other metabolic functions [69], an adequate supply of this vitamin is particularly important in a plant-based diet. This review revealed lower vitamin D intake in VGs in one study, but adequate levels were achieved through supplementation. Studies have shown that vitamin D intake is often too insufficient in children [70] and infants [71], so supplementation is generally recommended regardless of diet.

The vitamin B12 status was significantly lower in VGs and especially VEs than in OMs when not supplemented. Significantly reduced levels of vitamin B12 can lead to developmental delays and neurological deficits, particularly in children and adolescents [67]. It is important to consider whether or not vitamin B12 is being supplemented and, if so, by how much. According to a meta-analysis, vitamin B12 levels are significantly lower in VGs and VEs than in OMs. However, after subgroup analysis, this effect was no longer statistically significant in VGs, although it was in VEs. Those in the VE subgroup are particularly at risk of developing a vitamin B12 deficiency [67].

The mean daily calcium intake in milligrams did not differ significantly between VG and OM children across all studies in this review. As in a recent meta-analysis of adult data, the calcium intake of VEs in this review was the lowest compared to VGs and OMs; however, the differences between VGs and OMs were not significant [72]. Along with protein, calcium is considered an important nutrient for bone health [73]. Adequate calcium intake is particularly important during childhood and adolescence in order to achieve high peak bone mass. Meta-analyses of adults show reduced bone density in VGs and VEs and an increased fracture risk in VEs [59,74]. Comparable studies in children and adolescents are lacking, but differences in bone metabolism biomarkers have been observed (see ‘Health risk markers (inflammatory markers, blood lipids, bone markers’ in this review). Consistent with reviews and meta-analyses of adults [75] and vegan children [68], our review confirmed higher intakes of magnesium, vitamin C, vitamin E, and folate in VGs.

#### 4.2.6. Iron Intake and Status

Iron deficiency has been associated with negative effects on physical and mental development, particularly during childhood and adolescence [24,76]. Serum ferritin levels were significantly lower with VG compared to OM in three studies. An insufficient iron intake in VGs [75] could not be confirmed in our review.

### 4.3. Food Groups

Food consumption was assessed in only three studies. The lack of data on VG children’s dietary intake is also confirmed by other studies [77]. Children on VG diets consistently consumed higher amounts of food groups considered to be healthy [78,79], such as fruits, whole grains, nuts, and legumes. In contrast, sweets and sweetened beverages were consumed less by VGs, although the reasons for this are unclear. Either there are fewer vegetarian products without gelatine and fewer vegan products without eggs and milk, or the products are too expensive [23]. Alternatively, the improved health awareness of parents and older VG children may influence consumption [80]. It should be noted that a lower consumption of sugar before birth by the mother and afterwards can protect children from chronic diseases, so that a lower sugar consumption by VEs and VGs through fewer sweets or sweetened drinks can only be seen as positive. In three of the four studies, VGs consumed less milk and dairy products than OMs. This is confirmed in adult studies [81]. Milk and dairy products are often replaced by plant-based dairy products in VG diets. However, the nutrient profile differs from that of cow’s milk, and soy products in particular are considered critical due to their isoflavone content [23]. In adults, higher intakes of whole grains, vegetables, and nuts reduce the risk of chronic metabolic diseases [78]. These food groups were consumed more by VGs in our review (see ‘Food groups’ in this review).

### 4.4. Practical Implications

A physical examination or laboratory analysis of relevant biomarkers can provide valuable information for determining nutritional status and assessing age-appropriate physical development. A personalized nutrition plan that considers a VG or VE diet can help ensure the consumption of all necessary nutrients in the appropriate quantities. Parents may need more information. Pediatricians and general practitioners should offer advice on VG or VE diets for children and adolescents or be available to answer questions. They should also be sufficiently informed on the subject [14]. In Germany, for example, the Gemeinschaftskrankenhaus Herdecke offers a consultation hour for VE or VG children and their parents.

### 4.5. Limitations

Interpretation of the dietary intake results is limited by the different methods used to record intake and the inconsistent food group definitions. For example, the fruit food group was divided into fresh fruit or 100% fruit juice, canned fruit, and dried fruit [37]. Conversely, fresh, frozen, canned, and dried fruits were grouped together with 100% juices, smoothies, and squeezes [23]. In the review by Schürmann et al., the methods used to record and determine food consumption in the included studies already varied [1].

It is also important to consider that all studies included in the review included a wide range of age groups. For this reason, we performed a limited cumulative analysis, which allowed us to confirm some results and clarify others. A striking feature of the anthropometric data was the variety of recording methods and assessment measures used, which proved difficult to reconcile. Of the studies included in this systematic review, few had comparative data from the United States. This may be because the potential negative effects of a VG diet among children and young people are not critically examined in the United States [16], where there are generally significantly more vitamin- and mineral-fortified foods on the market than in Germany [82]. Therefore, it may be easier to provide the critical nutrients in these countries.

We found no further categorization of vegetarian diets in many of the included studies. Certainly, depending on the subcategory lacto-ovo, ovo, or pescetarian, the food and nutrient intake may again be different, and we cannot draw further conclusions as we did based on the studies roughly divided into VG and VE diets.

## 5. Conclusions

Finally, it should be noted that future studies should compare dietary intake and biomarker data with reference ranges to better assess deficiency risks or health outcomes [70]. The study design in almost all studies was cross-sectional, which does not allow conclusions to be drawn over time. Longitudinal, prospective observational studies are needed to strengthen the evidence for the effects of a vegetarian diet on dietary intake, health, and nutritional status in children and adolescents.

## Figures and Tables

**Figure 1 nutrients-17-02183-f001:**
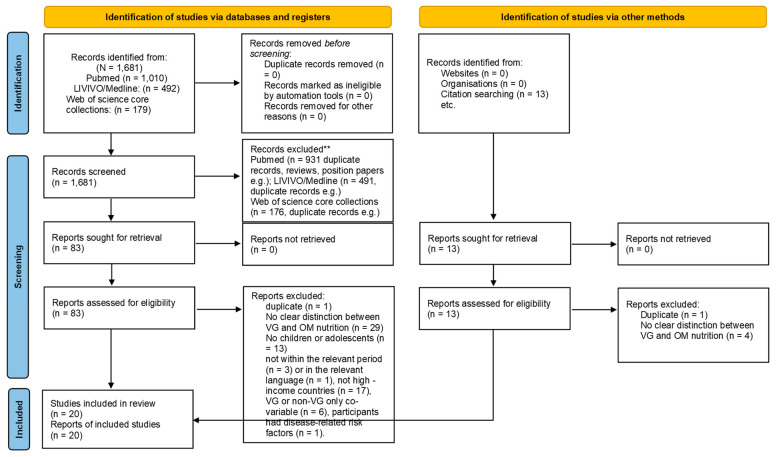
Prisma diagram inclusion process in the study. ** no automation tool was used.

**Figure 2 nutrients-17-02183-f002:**
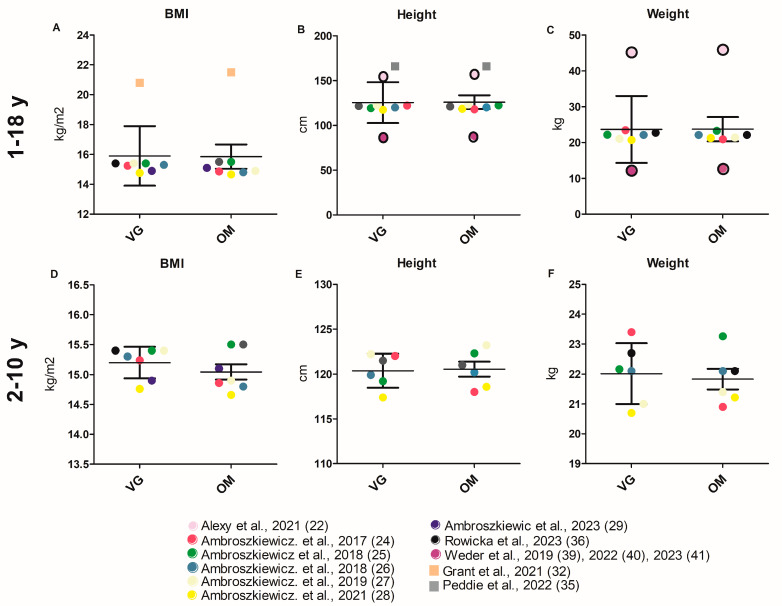
Anthropometric data of VG and OM children. Each data point corresponds to the mean value published in the indicated publication. (**A**–**C**) Studies involving all age groups (1–18 y) are included in the analysis. (**D**–**F**) Studies investigating children aged 2–10 years are included in the analysis. Significant difference between VG and OM group within a study are indicated by an asterisk (*) inside the data point. Only *p*-values of unadjusted datasets were considered. Data points for which no significance values (of unadjusted data) are available are framed in black. Square data points indicate that VEs are included in VG group [22,24,25,26,27,28,29,32,35,36,39,40,41].

**Figure 3 nutrients-17-02183-f003:**
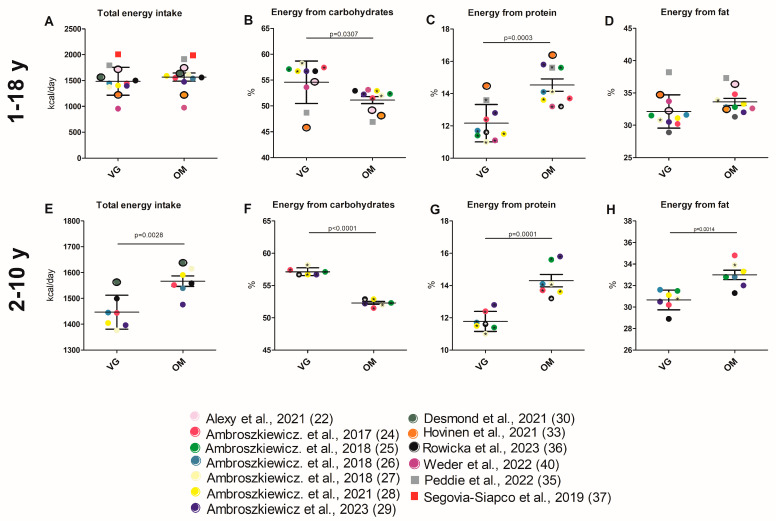
(**A**–**H**) Selected energy from nutrients of children with VG and OM diet. Each data point corresponds to the mean value published in the indicated publication. (**A**–**D**) Studies involving all age groups (1–18 y) are included in the analysis. (**E**–**H**) Studies investigating children aged 2–10 years are included in the analysis. Significant difference between VG and OM group within a study are indicated by an asterisk (*) inside the data point. Only *p*-values of unadjusted datasets were considered. Data points for which no significance values (of unadjusted data) are available are framed in black. Square data points indicate that VEs are included in VG group [22,24,25,26,27,28,29,30,33,35,36,37,40].

**Figure 4 nutrients-17-02183-f004:**
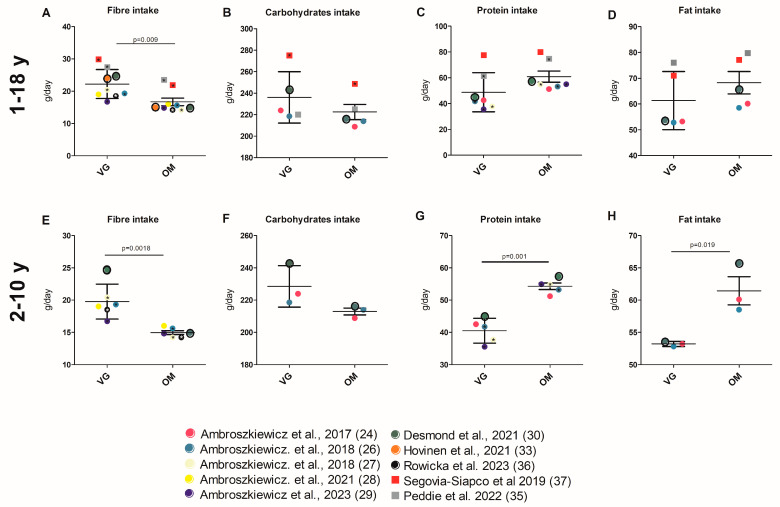
(**A**–**H**) Intake of selected nutrients of children with VG and OM diet. Each data point corresponds to the mean value published in the indicated publication. (**A**–**D**) Studies involving all age groups (1–18 y) are included in the analysis. (**E**–**H**) Studies investigating children aged 2–10 years are included in the analysis. Significant difference between VG and OM group within a study are indicated by an asterisk (*) inside the data point. Only *p*-values of unadjusted datasets were considered. Data points for which no significance values (of unadjusted data) are available are framed in black. Square data points indicate that VEs are included in VG group [24,26,27,28,29,30,33,35,36,37].

**Figure 5 nutrients-17-02183-f005:**
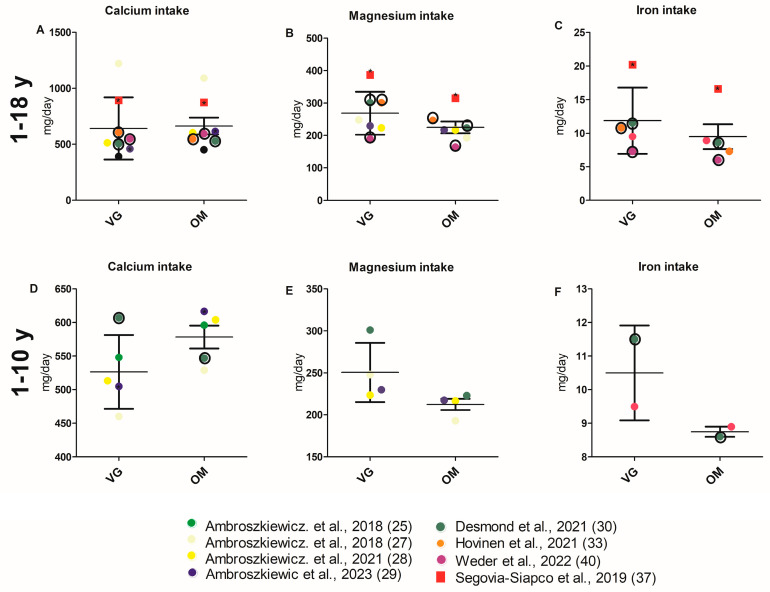
(**A**–**F**) Intake of selected minerals of children with VG and OM diet. Each data point corresponds to the mean value published in the indicated publication. (**A**–**C**) Studies involving all age groups (1–18 y) are included in the analysis. (**D**–**F**) Studies investigating children aged 2–10 years are included in the analysis. Significant difference between VG and OM group within a study are indicated by an asterisk (*) inside the data point. Only *p*-values of unadjusted datasets were considered. Data points for which no significance values (of unadjusted data) are available are framed in black. Square data points indicate that VEs are included in VG group [25,27,28,29,30,33,37,40].

**Figure 6 nutrients-17-02183-f006:**
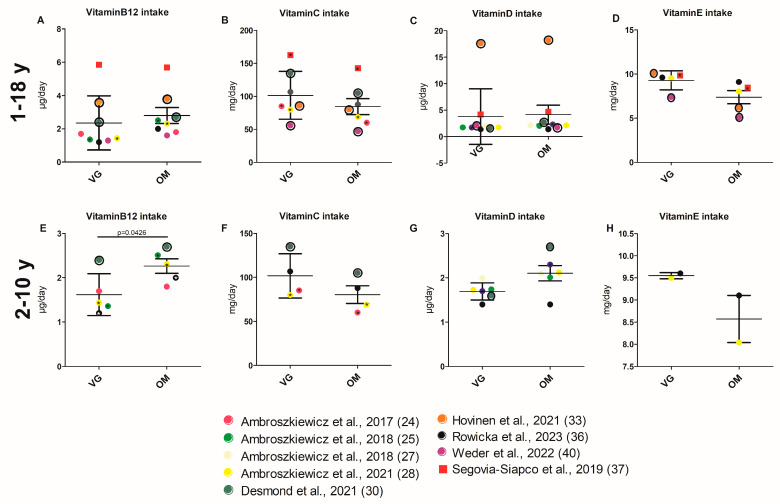
(**A**–**H**) Intake of selected vitamins of children with VG and OM diet. Each data point corresponds to the mean value published in the indicated publication. (**A**–**D**) Studies involving all age groups (1–18 y) are included in the analysis. (**E**–**H**) Studies investigating children aged 2–10 years are included in the analysis. Significant difference between VG and OM group within a study are indicated by an asterisk (*) inside the data point. Only *p*-values of unadjusted datasets were considered. Data points for which no significance values (of unadjusted data) are available are framed in black. Square data points indicate that VEs are included in VG group [24,25,27,28,30,33,36,37,40].

**Table 1 nutrients-17-02183-t001:** Characteristics and findings of the included studies.

Reference; Study Location (Implementation Period)	Study Design/ Study Quality ^1^ (Scale: 0–9)	Participant Characteristics	Assessed Dietary Intake Parameters	Assessed Biomarkers	Further Assessed Parameters	Sign. Diff. Between VG and OM/Main Outcome
Alexy et al., 2021 [22], Germany (2017–2018)	Cross-sectional study Subanalysis of the VeChi Youth Study ******	Total: N = 401 (229 f/172 m) Age group: 5–19 y VG: n = 150 (91 f/59 m) OM: n = 137 (62 f/75 m) Other groups: VE	Nutrients (total energy, protein, carbohydrates, sugar, fiber, fat) Vitamins (Vit.A, Vit.B1,2,9,12, Vit.C, Vit.E) Minerals (calcium, magnesium, iron, zinc)	Iron status (hemoglobin, ferritin) Vitamins (Vit.B2, folate, 25-OH vitamin D3, holotranscobalamin, methylmalonic acid) Blood lipids (triglycerides, total cholesterol, LDL, HDL)	Anthropometric data Sociodemographic data Physical activity Puberty status Dietary variables	VG ↑ Energy from carbohydrates (*p* = 0.0002) ^2^ ↑ Tocopherol-Equivalents intake (Vit.E) (*p* = 0.0015) ^2^ ↓ Intake of Vit.B1 (*p* = 0.0413) ^2^, Vit.B_2_ (*p* = 0.0149) ^2^, Vit.B12 (*p* = 0.0002) ^2^ ↓ Protein intake (*p* = 0.0011) ^2^ ↓ Fat intake (*p* = 0.0002) ↑ Fiber intake (*p* = 0.0006) ↓ Ferritin level (*p* = 0.0134) ^2^ ↓ Holotranscobalamin level (*p* = 0.0042) ^2^ ↑ Methylmalonic acid level (*p* = 0.0253) ^2^ Data show no nutritional risks for VG adolescents.
Alexy et al., 2022 [23], Germany, (2017–2018)	Cross-sectional study Subanalysis of the VeChi Youth Study *****	Total N = 390 (221 f/169 m) Age group: 5–19 y VG: n = 145 (87 f/58 m) OM: n = 135 (61 f/74 m) Other groups: VE	Food groups Nutrients (total energy, protein, fat, carbohydrates)	N/A	Anthropometric data Sociodemographic data Physical activity Dietary variables	VG ↓ Dairy intake (*p* = 0.0003) ↑ Grain intake (*p* = 0.0065) The lower intake of dairy products in VG children suggests the need for vitamin B12 supplementation.
Ambroszkiewicz et al., 2017 [24], Poland (2015–2016)	Cross-sectional study ******	Total: N = 89 (44 f/45 m) Age group: 4–9 y VG: n = 43 (25 f/18 m; mean age: 6.5 y) OM: n = 46 (19 f/27 m; mean age: 6.3 y)	Nutrients (total energy, protein, fat, carbohydrates) Vitamins (Vit.C, Vit.B12) Iron	Iron status (iron, hepcidin, sTfR, ferritin, transferrin, hemoglobin) Others (mean corpuscular volume, red blood cells, CRP)	Anthropometric data Sociodemographic data	VG ↓ Energy from protein (*p* = 0.03) ↑ Energy from carbohydrates (*p* = 0.001) ↑ Vit.C intake (*p* < 0.05) ↓ Ferritin level (*p* < 0.01) ↓ Hepcidin level (*p* < 0.05) ↑ sTfR level (*p* < 0.001) ↓ CRP (*p* = 0.011) sTfR and hepcidin are possible markers for the detection of subclinical iron deficiency in VG children.
Ambroszkiewicz, et al., 2018 [25], Poland (2014–2017)	Cross-sectional study ******	Total N = 130 Age group: 5–10 y VG: n = 70 (mean age: 6.6 y) OM: n = 60 (mean age: 6.9 y)	Nutrients (total energy, fat, carbohydrates, protein) Minerals (calcium, phosphorus) Vitamins (Vit.D, Vit.B12)	Bone metabolism markers (BALP, CTX-I, osteoprotegerin, RANKL, Sclerostin, Dkk-1)	Anthropometric data Sociodemographic data Body composition Bone mineral density (BMD) Bone mineral content (BMC)	VG ↓ Energy from protein (*p* < 0.001) ↑ Energy from carbohydrates (*p* = 0.004) ↓Vit.B12 intake (*p* < 0.001) ↑ BALP level (*p* = 0.002) ↑ CTX-I level (*p* = 0.027), positive correlation of CTX-I with BMC, total BMD, and lumbar spine BMD VG children do not have a lower bone mass; CTX-I might be an important marker for the protection of bone abnormalities in VG children.
Ambroszkiewicz et al., 2018 [26]; Poland (2017–2018)	Cross-sectional study ******	Total: N = 117 (56 f/61 m) Age group: 5–10 y VG: n = 62 (28 f/34 m; mean age: 6 y) OM: n = 55 (28 f/27 m; mean age: 6.5 y)	Nutrients (total energy, protein, carbohydrates, fiber, fat)	Adipokines (leptin/sOB-R, HMW/total adiponectin, resistin, visfatin, omentin, vaspin)	Anthropometric data Sociodemographic data	VG ↓ Fat free mass (*p* = 0.044) ↓ Energy from protein (*p* < 0.001) ↓ Protein intake (*p* = 0.002) ↑ Fiber intake (*p* = 0.011) ↓ Leptin/sOB-R ratio (*p* = 0.017) ↑ Ratios of anti-inflammatory to pro-inflammatory adipokines: adiponectin/leptin (*p* = 0.005); omentin/leptin (*p* = 0.011) The adipokine profile and inflammatory status of prepubertal children might be beneficially affected by VG diet.
Ambroszkiewicz et al., 2019 [27], Poland (2014–2016)	Cross-sectional study ******	Total: N = 106 Age group: 5–10 y VG: N = 53 (median age: 7.0 y) OM: n = 53 (median age: 7.0 y)	Nutrients (total energy, protein, fat, carbohydrates, fiber) Minerals (calcium, phosphorus, magnesium) Vit.D	Adipokines (leptin, adiponectin) Bone metabolism markers (CTX, osteocalcin (OC), CICP, parathormone, 25 hydroxyvitamin D)	Anthropometric data Sociodemographic data Bone mineral density	VG ↓ Fat mass (*p* = 0.018) ↓ TBLH-BMD z-score (*p* = 0.009); BMDL2-L4 z-score (*p* = 0.019) ↓ Leptin level (*p* < 0.001); Leptin/adiponectin ratio (*p* < 0.001) ↓ OC/CTX ratio 0.039 ↑ c-OC/uc-OC ratio (*p* = 0.010); c-OC/OC ratio (*p* = 0.048) ↑ Parathormone level (*p* = 0.015) ↓ Energy from protein (*p* = 0.001), protein intake (*p* = 0.002) ↓ Energy from fat (*p* = 0.043) ↑ Energy from carbohydrates (*p* = 0.001) ↑ Fiber intake (*p* = 0.015) Values of bone mineral density are similar in VG and OM children. Significantly lower total and lumbar spine BMD z-scores were seen in VG children.
Ambroszkiewicz et al., 2021 [28], Poland (2018–2020)	Cross-sectional study ******	Total N = 105 (52 f/53 m) Age group: 5–9 y VG: n = 55 (26 f/29 m; mean age: 5.5 y) OM n = 50 (26 f/24 m; mean age: 6.0 y)	Nutrients (total energy, fiber, fat, carbohydrates, protein) Vitamins (Vit.A, Vit B12, Vit.C, Vit.D, Vit.E) Minerals (calcium, magnesium, manganese, phosphorus)	Adipokines (Adiponectin, Visfatin, Omentin, Leptin) Myokines (Myostatin, Irisin)	Anthropometric data Sociodemographic data	VG ↓ Fat mass (*p* = 0.018) ↓ Leptin level (*p* = 0.003) ↓ Energy from protein (*p* < 0.001) ↑ Energy from carbohydrates (*p* = 0.002) ↑ Fiber intake (*p* = 0.063) ↑ Manganese intake (*p* = 0.020) ↓ Vit.B12 intake (*p* < 0.001) ↑ Vit.C intake (*p* = 0.019) A lacto-ovo vegetarian diet has no impact on myokines and adipokines levels in prepubertal children.
Ambroszkiewicz et al., 2023 [29], Poland (2020–2021)	Cross-sectional study ******	Total N = 76 (37 f/39 m) Age group: 5–9 y VG: n = 51 (25 f/26 m; mean age: 6 y) OM: n = 25 (12 f/13 m; mean age: 5.5 y)	Nutrients (total energy, protein, fiber, fat, carbohydrates, protein) Vit.D Minerals (calcium, magnesium, phosphorus) Amino acids (methionine, phenylalanine, histidine, threonine, tryptophan, valine, isoleucine, leucine, lysine, alanine, arginine, aspartate, glutamate, glycine, proline, serine, cysteine, tyrosine)	Amino acids (methionine, phenylalanine, histidine, threonine, tryptophan, valine, isoleucine, leucine, lysine, alanine, arginine, aspartate, glutamate, glycine, proline, serine, cysteine, tyrosine, ornithine, citrulline, taurine) Albumin, prealbumin Bone metabolism markers (25-hydroxyvitamin D, IGF-I, CTX-I, parathormone, osteocalcin, osteoprotegerin)	Anthropometric data Sociodemographic data	VG ↓ Energy from protein (*p* = 0.0002); protein intake (*p* = 0.0009) ↑ Energy from carbohydrates (*p* = 0.0125) ↑ Fiber intake (*p* = 0.0034) ↓ Calcium intake (*p* = 0.0149) ↓ Vit.D intake (*p* = 0.0116) ↓ Amino acids intake (*p* < 0.001) ↓ Valine level (*p* = 0.0253), Lysine level (*p* = 0.0297), Leucine level (*p*= 0.0315), Isoleucine level (*p* = 0.0231) ↓ Albumin level (*p* = 0.0001) ↑ CTX-I level (*p* = 0.0343) Significant relationships of osteoprotegerin with alanine, ornithine, and aspartate might suggest an impact of diet on the bone regulatory pathway.
Desmond et al., 2021 [30], Poland (2014–2016)	Cross-sectional study ******	Total: N = 187 (100 f/87 m) Age group: 5–10 y VG: n = 63 (32 f/31 m; mean age: 7.6 y) OM: n = 72 (38 f/34 m; mean age: 7.7 y) Other groups: VE	Nutrients (total energy, carbohydrates, starch, sucrose, fiber, fat, cholesterol) Vitamins (beta carotene equivalents, Vit.B_12_, folate, Vit.C, Vit.D) Minerals (calcium, magnesium, iron)	Cardiovascular risk marker (Insulin, fasting glucose, HOMA-IR, total cholesterol, HDL, LDL, VLDL, triglycerides, hs-CRP, cIMT, IGFBP-3, IGF-1) Iron status (red blood cells, hemoglobin, hematocrit, ferritin) Vit.B12	Anthropometric data Sociodemographic data Physical activity Body composition Bone mineral density Bone mineral content (BMC) Dietary variables	↓ Protein intake, sucrose intake, fat intake, cholesterol intake, Vit.B_12_ intake (without supplementation), Vit.D intake (with and without supplementation) (*p* < 0.001) ^3^ ↑ Carbohydrate intake, starch intake, fiber intake, folate intake, beta-carotene intake, Vit.C intake, magnesium intake, calcium intake, iron intake (*p* < 0.001) ^3^ ↓ Height z score (*p* < 0.05) ^3^ ↓ Thigh girth z score (*p* < 0.05) ^3^ ↓ Total body less head BMC (*p* < 0.01) ^3^ ↑ Fasting glucose level (*p* < 0.01) ^3^ ↓ HDL cholesterol level (*p* < 0.05) ^3^ ↓ VLDL cholesterol level (*p* < 0.05) ^3^ ↑ Triglycerides level (*p* < 0.01) ^3^ ↓ Hematocrit (*p* < 0.05) ^3^ ↓ Ferritin level (*p* < 0.05) ^3^ VGs showed fewer nutritional deficiencies but a more unfavorable cardiometabolic risk profile than OMs.
Elliott et al., 2022 [31] Canada (2008–2019)	Longitudinal cohort study ********	Total: N = 8907 (4242 f/4665 m) Age group: 0.5–8 y VG(+VE): n = 248 (111 f/137 m; mean age: 2.3 y) OM: n = 8659 (4131 f/4528 m; mean age: 2.2 y)	N/A	Iron status (ferritin, 25-hydroxyvitamin D 25[OH]D) Blood lipids (total cholesterol, HDL, LDL, triglycerides)	Anthropometric data Sociodemographic data Dietary variables	VG: ↓ Height z score (*p* < 0.02) ↑ Underweight (*p* = 0.008) Growth or biochemical measures of nutrition do not meaningfully differ between VG children and OM children. However, VG diet is associated with a higher odd of underweight.
Grant et al., 2021 [32], Australia (2008)	Cross-sectional study ******	Total: N = 688 (602 f/83 m) Age group: 14–17 y VG: n = 49 (39 f/9 m) OM: n = 639 (563 f/74 m)	N/A	Cardiovascular risk marker (CRP, total cholesterol, HDL, LDL, triglycerides, glucose)	Anthropometric data Sociodemographic data Blood pressure (BP)	VG: ↓ Total and LDL-cholesterol level (*p* = 0.001) ↑ Diastolic BP (*p* = 0.038) The prevalence rate of participants with 3 or more risk factors was similar among VG and OM participants. Abnormal cholesterol values were detected in both diet groups.
Hovinen et al., 2021 [33], Finland (2017)	Cross-sectional study ******	Total: 40 (19 f/21 m) Age group: 1–7 y VG: n = 10 (4 f/6 m; mean age: 3.37 y) OM: n = 24 (12 f/12 m; mean age: 3.89 y) Other groups: VE	Nutrients (total energy, sucrose, fiber, fat, carbohydrates, protein, cholesterol) Vitamins (Vit.A, thiamine (B1), riboflavin (B2), niacin equivalents (B3), pyridoxine (B6), cobalamin (B12), folate (B9), Vit.C, Vit.D, Vit.E, Vit.K) Minerals (calcium, magnesium, iron, sodium, potassium, zinc, iodine, phosphorus) Others (EPA, DHA)	Vitamins (RBP, Vit.A Folate, Vit.B12, Vit.D3, Vit.D2) Minerals (zinc, iodine) Cholesterol metabolism (e.g., cholesterol, LDL, HDL, cholestanol) Primary, secondary bile acids and biosynthesis marker (e.g., cholic, lithocholic acid, hyodeoxycholic) Inflammatory marker (CRP, ambulatory glucose profile) Others (transthyretin, ferritin, TfR, glucoses)	Anthropometric data Sociodemographic data	VG: ↓ Energy from saturated fatty acids (*p* = 0.029) ^4^ ↑ Energy from polyunsaturated fatty acid (*p* = 0.0027) ^4^ ↑ Linoleic acid (*p* = 0.0037) ^4^, alpha-linolenic acid intake (*p* = 0.0057) ^4^ ↓ Cholesterol intake (*p* = 0.034) ^4^ ↑ Fiber intake (*p* = 0.0036) ^4^ ↑ Thiamine (B1) intake (*p* = 0.034) ^4^; Folate (B9) ^4^ intake (*p* = 0.0034) ^4^ ↑ Magnesium intake (*p* = 0.034) ^4^, iron intake (*p* = 0.0038) ^4^ ↓ Zinc level (*p* = 0.039) ^4^ ↑ Sitosterol level (*p* = 0.021) ^4^ ↑ Avenasterol level (*p* = 0.021) ^4^ ↑ Cholestenol level (*p* = 0.021) ^4^ ↑ Lathosterol level (*p* = 0.021) ^4^ The risk of nutrient deficiency in children can be reduced by part-time consumption of lacto-ovo-vegetarian products in an otherwise strict vegan diet.
Nieczuja-Dwojacka et al., 2020 [34], Poland (2015–2016)	Cross-sectional study ******	Total: N = 218 (100 f/118 m) Age group: 3–15 y VG: n = 47 (22 f/25 m) OM: n = 171 (78 f/93 m)	N/A	N/A	Anthropometric data Sociodemographic data Reaction time	VG: ↓ Height (*p* < 0.05) ↓ BMI (*p* < 0.01) ↓ Sum of three skinfolds (*p* < 0.01) ↑ Reaction time (*p* < 0.05) VG diet affects the height, BMI, and body fatness, as well as the reaction time.
Peddie et al., 2022 [35], New Zealand (2019)	Cross-sectional study *****	Total: N = 254 f, Age group: 15–18 y VG(+VE): n = 38 f (mean age 17.1 y) OM: n = 216 f (mean age 16.8 y)	Food groups Nutrients (total energy, protein, carbohydrates, sugar, fiber, fat)	N/A	Anthropometric data Sociodemographic data	VG: ↓ BMI z-score (*p* = 0.003) ↓ Consumers of poultry, sausages and processed meat; red meat; eggs and egg-based dishes; pies and pasties (*p* < 0.05) ↑ Consumers of vegetarian meat alternatives (*p* < 0.05) ↑ Energy intake from legumes (*p* = 0.011) and vegetables (*p* = 0.012) ↓ Protein intake (*p* < 0.001) ↓ Saturated fat intake (*p* = 0.014) ↑ Energy from polysaturated fat (*p* < 0.001) ↑ Energy from fiber intake (*p* = 0.019) VG children consume more fiber, more polyunsaturated fat, and less protein than children with an OM diet. Some VGs consume food groups associated with poorer dietary quality.
Rowicka et al., 2023 [36] Poland (2020–2021)	Cross-sectional study ******	Total N = 72 (39 f/33 m) Age group: 2–10 y VG: n = 32 (17 f/15 m) OM: n = 40 (22 f/18 m)	Nutrients (total energy, fiber, fat, protein, carbohydrates) Vitamins (Vit.A, Vit.B12, Vit.C, Vit.D, Vit.E)	Oxidant–antioxidant marker (total antioxidant capacity (TAC), total oxidant capacity (TOC), reduced glutathione (GSH), oxidized glutathione (GSSG)) Others (CRP, calprotectin)	Anthropometric data Sociodemographic data Pubertal stage	VG: ↓ Energy from protein (*p* = 0.010) ↑ Energy from carbohydrates (*p* = 0.005) ↑ Fiber intake (*p* = 0.012) ↓ Vit.B12 (*p* < 0.0001) ↓ TOC (*p* = 0.001), GSH level (*p* = 0.001), GSSG level (*p* = 0.002) ↑ TAC (*p* < 0.001) ↓ Oxidative stress index (*p* < 0.001) Maintaining the oxidant–antioxidant balance in prepubertal VG children is possible.
Segovia-Siapco et al., 2019 [37], USA (-)	Cross-sectional study; Subanalysis of the Teen Food and Development Study ****	Total N = 534 (302 f/323 m) Age group: 12–18 y; VG(+VE): n = 137 (90 f/47 m) OM: n = 397 (212 f/185 m)	Food groups Nutrients (total energy, protein, carbohydrates, sugar, fiber, fat) Vitamins (thiamine, riboflavin, Vit.B_12_, Folate, Vit.C, Vit.D., Vit.E) Minerals (magnesium, iron, sodium, potassium, zinc)	N/A	Anthropometric data Sociodemographic data Physical activity Duration of sleep	VG: ↑ Consumption of breads/grains/pastas/cereals (*p* = 0.022), fruits (*p* = 0.001), vegetables (*p* < 0.0001), nuts, nut butters, meat alternatives (*p* < 0.0001), dairy substitutes (*p* < 0.0001) ↓ Consumption of meat, poultry, eggs (*p* < 0.0001), cheese, dairy (*p* = 0.003), milk, dairy (*p* < 0.0001), dairy desserts (*p* = 0.007), sugar-sweetened beverages (*p* <0.0001), coffee/tea (*p* = 0.002) ↑ Intake of carbohydrates, fiber, thiamin, Vit.E, folate, calcium, iron, potassium, magnesium (*p* < 0.0001), sodium (*p* = 0.022), Vit.C (*p* = 0.029) ↓ Intakes of fat, zinc (*p* < 0.0001) VG adolescents have a more favorable dietary intake profile.
Světnička et al., 2022 [38] Czech Republic (2019–2021)	Cross-sectional study ******	Total N = 200 (100 f/100 m) Age group: 0–18 y VG n = 79 (44 f/35 m; median age: 4.5 y) OM: n = 52 (25 f/27 m; median age: 4.5 y) Other groups: VE	Vit.B_12_	Cyanocobalamin (Vit.B12) Holotranscobalamin Hemoglobin Homocysteine Mean corpuscular volume Folate	Anthropometric data Sociodemographic data Dietary variables	↑ Cyanocobalamin (B12) level (*p* = 0.019) ^5^ VG diet does not cause severe vitamin B12 deficiency but rather supplementation seems to lead to vitamin B12 hypervitaminosis in several of the investigated children.
Weder et al., 2019 [39], Germany (2016 –2018)	Cross-sectional study Subanalyses of the VeChi-Diet-Study ******	Total: N = 430 (223 f/207 m) Age group: 1–3 y VG: n = 127 (64 f/63 m, mean age: 2 y) OM: n = 164 (83 f/81 m; mean age: 2 y) Other groups: VE	Nutrients (total energy, fat, sugar, fiber, carbohydrates, protein) Others (EPA, DHA)	N/A	Anthropometric data Sociodemographic data Physical activity	↓ Protein intake (*p* ≤ 0.001) ^6^ ↑ Fiber intake (*p* ≤ 0.001) ^7^ Children with VG and OM diet in early childhood have the same amount of energy and macronutrients intake, leading to a normal growth in comparison to OM children.
Weder et al., 2022 [40], Germany (2016–2018)	Cross-sectional study Subanalyses of the VeChi-Diet-Study ******	Total: N = 430 (223 f/207 m) Age group: 1–3 y VG: n = 127 (64 f/63 m, mean age: 2 y) OM: n = 164 (83 f/81 m; mean age: 2 y) Other groups: VE	Nutrients (total energy, carbohydrates, protein, fat, cholesterol) Vitamins (Vit.A, beta Carotin, Vit.E, Vit.K, Vit.B_1,2,6,12_, folate, Vit.C, Vit.D) Minerals (potassium, calcium, magnesium, iron, zinc, iodine)	N/A	Anthropometric data Sociodemographic data Physical activity Dietary variables	↓ Protein intake, Energy from protein (*p* ≤ 0.001) ↑ Vit.E intake (*p* ≤ 0.001) ^8^ ↓ Vit.B2 intake (*p* ≤ 0.001) ^9^ ↑ Magnesium ^10^, iron ^11^ intake (*p* ≤ 0.001) ↓ DHA, EPA intake (*p* ≤ 0.001) ^12^ VG diets can provide most micronutrients in desirable amounts and a preferable fat quality in young children (1–3 y) compared to an OM diet.
Weder et al., 2023 [41] Germany (2016–2018)	Cross-sectional study Subanalyses of the VeChi Diet Study ******	Total: N = 430 (223 f/207 m) Age group: 1–3 y VG: n = 127 (64 f/63 m, mean age: 2 y) OM: n = 164 (83 f/81 m; mean age: 2 y) Other groups: VE	Selenium	N/A	N/A	VGs have a lower selenium consumption (not significant) than OMs. A total of 39% of VG and 16% of OM children consume less than the recommended amount of selenium.

^1^: Quality of studies was assessed by using the NOS scale. A study can be awarded a maximum of one star for each item within the Selection (4 items) and Outcome (3 items) categories. A maximum of two stars can be given for Comparability. ^2^: Data were adjusted for age, BMI-SDS, socioeconomic status, smoking in the household, physical activity, use of dietary supplements; *p*-values were adjusted for multiple testing according to the False Discovery Rate (FDR) method. ^3^: Data were adjusted to diet group, age, sex. ^4^: *p*-values were calculated using age- and sex-adjusted exact permutation tests with n = 47,500 permutations and Benjamini–Hochberg correction for multiple testing. ^5^: Parametric analysis of variance (MCV) or Kruskal–Wallis test was performed to detect differences between VGs, VEs, and OMs. ^6^: Data were adjusted for age, sex, breast milk intake, SES, weight-for-height z-score. ^7^: Data were adjusted for age, sex, breast milk intake, SES, weight-for-height z-score, urbanicity. ^8^: Data were adjusted for age, sex, breast milk intake, weight-for-height *z*-score, SES, seasons. ^9^: Data were adjusted for age, sex, breast milk intake, SES, energy intake, paternal BMI. ^10^: Data were adjusted for age, sex, breast milk intake, energy intake, weight-for-height z-score, paternal BMI, seasons. ^11^: Data were adjusted for age, sex, breast milk intake, energy intake, weight-for-height z-score, physical activity, SES, seasons, urbanicity. ^12^: Data were adjusted for age, sex, breast milk intake, physical activity, energy intake, seasons. BALP: bone alkaline phosphatase; BMDL2-L4 e lumbar spine L2eL4 bone mineral density; CICP e carboxy-terminal propeptide of type I collagen; cIMT: carotid intima media thickness; (hs-) CRP: high sensitive *C*-reactive protein; CTX-I: *C*-telopeptide of crosslinked collagen type I; DHA: docosahexaenoic acid; Dkk-1: Dickkopf-related protein 1; EPA: eicosapentaenoic acid; HMW: high-molecular-weight adiponectin; HOMA-IR: homeostatic model assessment for insulin resistance; HDL: high-density lipoprotein; IGF-1: insulin-like growth factor 1; IGFBP-3: insulin-like growth factor-binding protein 3; LDL: low-density lipoprotein; RBP: retinol-binding protein; RANKL: receptor activator of nuclear factor kappa-Β ligand; sTfR: soluble transferrin receptor; sOB-R: soluble leptin receptor; TBLH: total bone mineral density less head; VLDL: very low-density lipoprotein. ↑ VG values are higher than OM values. ↓ VG values are lower than OM values.

## Data Availability

Detailed analyses can be found in the Appendix A.

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
