# Peer review of "Vegetarian Diet and Dietary Intake, Health, and Nutritional Status in Infants, Children, and Adolescents: A Systematic Review"

_nutrients, 2025, doi:10.3390/nu17132183_

Round 1
Reviewer 1 Report
Comments and Suggestions for Authors
This review addresses an important topic, but several key revisions are needed. The conclusions are repeatedly described as inconclusive, which weakens the manuscript. A more focused synthesis or limited meta-analysis on consistent outcomes would improve clarity. The broad age range of 0 to 18 years is treated without stratification, despite differing nutritional needs across ages. Age-specific analysis would enhance interpretability. Although the Newcastle-Ottawa Scale is mentioned, no summary of study quality is provided; a brief table or narrative would improve transparency.
Minor issues include inconsistent terminology and grammatical errors that affect readability, and the figure legends lack clarity, particularly regarding the inclusion of vegans. Supplementary tables are cited but not fully explained in the text. The manuscript would also benefit from discussing practical implications, such as the need for pediatric monitoring or supplementation in vegetarian children. Addressing these points would strengthen both the scientific rigor and applicability of the review.
Author Response
Dear reviewer 1,
Thank you for your review. We are pleased to hear that our review addresses an important topic. We would like to respond to your comments below. We have made some changes based on your comments as corrections in the manuscript. For others, we would like to explain our point of view.
Comment 1.) The conclusions are repeatedly described as inconclusive, which weakens the manuscript. A more focused synthesis or limited meta-analysis on consistent outcomes would improve clarity. The broad age range of 0 to 18 years is treated without stratification, despite differing nutritional needs across ages. Age-specific analysis would enhance interpretability.
Due to the heterogeneous ages of the study participants, we additionally conducted a limited cumulative analysis of stratification by age category. We focused on the 2–10 age group in the cumulative analysis, excluding studies that examined participants of different ages. Most of these studies focused on adolescents. We addressed this in the Methods, Results, and Discussion sections.
Comment 2.) Although the Newcastle-Ottawa Scale is mentioned, no summary of study quality is provided; a brief table or narrative would improve transparency.
We added an Excel file (Supplement 1: Studies Risk of Bias and Certainty of Evidence) to the supplements. This file detailss how we evaluated the quality of the studies. We uploaded the file again as a PDF and deleted the Excel version.
Comment 3.) Minor issues include inconsistent terminology and grammatical errors that affect readability, and the figure legends lack clarity, particularly regarding the inclusion of vegans.
We have reviewed the manuscript several times in an effort to correct any inconsistent terminology and grammatical errors. The square data points indicate that the VEs are included in the VG group. This is explained in the figure legends.
Comment 4.) Supplementary tables are cited but not fully explained in the text.
We explained the contents of the supplemental tables in the text.
Comment 5.) The manuscript would also benefit from discussing practical implications, such as the need for pediatric monitoring or supplementation in vegetarian children. Addressing these points would strengthen both the scientific rigor and applicability of the review.
We have considered your helpful comment under section 4.3, "Practical Implications
Reviewer 2 Report
Comments and Suggestions for Authors
The manuscript does a comprehensive review of the evidence regarding vegetarian diet and nutritional and health-related outcomes in children. The methodology is clearly explicated and the results are supported by the analysis. I have just few suggestions:
- The figures 2 and 3 have text that is difficult to read. I maybe better of look for alternatives to make them easy to read.
- It is not clear to me why include articles in German (an academic reason, more than the country of origen of the authors). Someone can say why do not include articles in French? or Polish (since large amount of selected evidence come from Poland). I suggest to explain this.
- I am curious how can be explained that few articles were selected from the US. I maybe interesting to add some discussion on this. Maybe little interest on this topic in the US?
Other than that, I believe that the manuscript is well written, make a contribution.
Author Response
Dear reviewer 2,
Thank you for your review. We are very pleased that the manuscript provided you with a comprehensive overview and that you have only a few suggestions for improvement. We would like to respond to your comments below. We have made some changes based on your comments as corrections in the manuscript. For others, we would like to explain our position.
Comment 1.) The figures 2 and 3 have text that is difficult to read. I maybe better of look for alternatives to make them easy to read.
To improve readability, the text in the illustrations has been enlarged.
Comment 2.) It is not clear to me why include articles in German (an academic reason, more than the country of origen of the authors). Someone can say why do not include articles in French? or Polish (since large amount of selected evidence come from Poland). I suggest to explain this.
We excluded the two articles in German and included two articles in English.
Comment 3.) I am curious how can be explained that few articles were selected from the US. I maybe interesting to add some discussion on this. Maybe little interest on this topic in the US?
We have considered this helpful comment in the limitations section, providing a justification.
Round 2
Reviewer 1 Report
Comments and Suggestions for Authors
This is a comprehensive and timely systematic review that adheres well to PRISMA guidelines. The authors provide a clear research objective, apply rigorous inclusion/exclusion criteria, and synthesize findings across a diverse set of studies with appropriate methodological transparency.
The manuscript presents a balanced evaluation of the potential nutritional benefits and risks associated with vegetarian diets in children and adolescents. The discussion is supported by a wide array of anthropometric, dietary, and biomarker data, and the cumulative analyses add valuable insight, particularly for the 2–10 year age group.
I find the work to be of high scientific quality, and the manuscript is well-structured, informative, and relevant to the current literature.
I recommend this manuscript for acceptance in its current form.